# Role of TNF-α-Inducing Protein Secreted by *Helicobacter pylori* as a Tumor Promoter in Gastric Cancer and Emerging Preventive Strategies

**DOI:** 10.3390/toxins13030181

**Published:** 2021-03-01

**Authors:** Masami Suganuma, Tatsuro Watanabe, Eisaburo Sueoka, In Kyoung Lim, Hirota Fujiki

**Affiliations:** 1Graduate School of Science and Engineering, Saitama University, Saitama 338-8570, Japan; 2Department of Drug Discovery and Biomedical Sciences, Faculty of Medicine, Saga University, Nabeshima, Saga 849-8501, Japan; sn6538@cc.saga-u.ac.jp; 3Department of Clinical Laboratory Medicine, Faculty of Medicine, Saga University, Nabeshima, Saga 849-8501, Japan; sueokae@cc.saga-u.ac.jp (E.S.); uv4h-fjk@asahi-net.or.jp (H.F.); 4Department of Biochemistry and Molecular Biology, Ajou University School of Medicine, Suwon 16499, Gyeonggi-do, Korea; iklim@ajou.ac.kr

**Keywords:** EMT, gastric cancer, nucleolin, HP-MP1, Tipα, TNF-α

## Abstract

The tumor necrosis factor-α (TNF-α)-inducing protein *(tipα)* gene family, comprising *Helicobacter pylori* membrane protein 1 *(hp-mp1)* and *tipα*, has been identified as a tumor promoter, contributing to *H. pylori* carcinogenicity. Tipα is a unique *H. pylori* protein with no similarity to other pathogenicity factors, CagA, VacA, and urease. American *H. pylori* strains cause human gastric cancer, whereas African strains cause gastritis. The presence of Tipα in American and Euro-Asian strains suggests its involvement in human gastric cancer development. Tipα secreted from *H. pylori* stimulates gastric cancer development by inducing TNF-α, an endogenous tumor promoter, through its interaction with nucleolin, a Tipα receptor. This review covers the following topics: tumor-promoting activity of the Tipα family members HP-MP1 and Tipα, the mechanism underlying this activity of Tipα via binding to the cell-surface receptor, nucleolin, the crystal structure of rdel-Tipα and N-terminal truncated rTipα, inhibition of Tipα-associated gastric carcinogenesis by tumor suppressor B-cell translocation gene 2 (*BTG2^/TIS21^*), and new strategies to prevent and treat gastric cancer. Thus, Tipα contributes to the carcinogenicity of *H. pylori* by a mechanism that differs from those of CagA and VacA.

## 1. Introduction

*Helicobacter pylori* is a Gram-negative bacterium that resides in the gastric lumen and is an important human pathogen. In 1984, Marshall and Warren published their findings on the association of *H. pylori* infection with chronic gastritis and peptic ulcers [1]. *H. pylori* was later classified as a definitive carcinogen in humans (Class 1) based on epidemiological studies [2]. The complete genome sequence of *H. pylori* strain 26695, which comprises 1,667,867 base pairs, harbors intricate systems responsible for motility, iron uptake, and DNA restriction and modification [3]. Infection with *H. pylori* strains induced gastric cancer in Mongolian gerbils [4,5]. Administration of *N*-methyl-*N*’-nitro-*N*-nitrosoguanidine (MNNG), as a tumor initiator, followed by *H. pylori* inoculation, as a tumor promoter, induced adenocarcinoma in the glandular stomach of Mongolian gerbils [6]. The results showed that *H. pylori* exhibits both tumor-initiating and -promoting activities. Epidemiological studies have revealed that *H. pylori* is relatively non-malignant in individuals of African ancestry, whereas it is harmful in humans of American ancestry [7,8]. The complete genome sequence of 60 *H. pylori* strains were assigned to populations from America, Africa1, Africa2, Euro-Asia, and East Asia. The vacuolating cytotoxin autotransporter (*vacA*) gene was identified in Euro-Asian and Africa1 strains. Several *vacA*-like genes were found in Africa2 strains, but not in East Asian and American populations [9]. The prevalence of cytotoxin-associated gene A (*cagA*)-positive strains is associated with a high incidence of atrophic gastritis and gastric cancer in Japanese and Korean populations [10,11]. In addition, the *cag* pathogenicity island (*cag* PAI) is present in 100% of East Asian and 60-70% of Western strains [12]. However, the genetic diversity of *H. pylori* suggests the existence of other genes and proteins that contribute to the development of human gastric cancer [13]. This review summarizes a new gene family, tumor necrosis factor-α (TNF-α)-inducing protein *(tipα),* which is secreted by *H. pylori* and acts as a tumor promoter. The *tipα* gene family includes *tipα*, *H. pylori* membrane protein 1 (*hp-mp1*), and possibly *jhp0543* [14].

The concept of tumor promoters originated from the “Reiztheorie” (inflammation theory) established by Rudolf Virchow [15]. Virchow reported the significance of chronic inflammation as a common factor in carcinogenesis. The experimental model of the two-stage chemical carcinogenesis process, consisting of initiation with a limited amount of 7,12-dimethybenz(a)anthracene (DMBA) and tumor promotion with 12-*O*-tetradecanoylphorbol-13-acetate (TPA) on mouse skin showed the significant role of inflammation in cancer development [16]. Tumor promoters, such as TPA and okadaic acid, commonly induce TNF-α, a pro-inflammatory cytokine, in their target organs. However, their mechanisms of action are different: TPA activates protein kinase C, and okadaic acid inhibits protein phosphatase 1 and 2A [17,18]. Furthermore, a two-stage carcinogenesis experiment using TNF-α-knockout mice revealed the key role of TNF-α in tumor-promoting inflammation. Treatment with DMBA plus okadaic acid did not result in tumors on the backs of TNF-α^−/−^ 129/svj mice for up to 19 weeks. Tumors developed in only 10% of the mice after 20 weeks, although the treatment induced tumors in 100% of TNF-α^+/+^ CD-1 mice and TNF-α^+/+^ 129/svj mice [19]. Thus, TNF-α plays the role of an endogenous tumor promoter, and chemical tumor promoters are inducers of TNF-α in their target organs. Okadaic acid stimulated gastric cancer development in rats treated with MNNG, along with strong induction of *TNF-α* in the stomach [18]. Thus, strong inducers of TNF-α in *H. pylori* act as tumor promoters in *H. pylori*-induced gastric cancer. 

The Tipα family (HP-MP1 and Tipα) is structurally and functionally unrelated to *H. pylori* pathogenicity factors such as VacA, CagA, and urease. HP-MP1 from *H. pylori* strain SR 7791, a 16 kDa protein, induces the secretion of various pro-inflammatory cytokines such as TNF-α, interleukin-1α (IL-1α), and IL-8 from human monocytes [20]. Transfection of *HP-MP1* into v-Ha-*ras*-transfected BALB/3T3 cells (Bhas 42 cells) induced strong *tnf-α* gene expression and produced tumors in nude mice [21]. *HP0596* of *H. pylori* strain 26695 [3] showed 94.3% homology to *hp-mp1* and was designated as Tipα due to its strong TNF-α-inducing activity [22]. Tipα is a key tumor promoter associated with *H. pylori* infection and human gastric cancer. This concept was supported by a report from Montano et al. [9], in which Tipα was detected in Euro-Asian and American *H. pylori* strains, the latter of which is a malignant *H. pylori* strain and does not harbor the *vacA* gene. 

As the potential pathogenic significance of Tipα in *H. pylori* strains worldwide is garnering attention, this review covers the following pertinent topics: (1) tumor-promoting activities of the members of the Tipα family, HP-MP1 and Tipα; (2) secretion of Tipα by *H. pylori*; (3) cellular response to recombinant Tipα (rTipα); (4) crystal structure of rdel-Tipα and N-terminal truncated rTipα; (5) nucleolin as a cell-surface receptor for rTipα; (6) epithelial–mesenchymal transition (EMT) induced by rTipα; (7) inhibition of Tipα-associated gastric carcinogenesis by B-cell translocation gene 2 (*BTG2^/TIS21^*), a tumor suppressor gene; and (8) new strategies to prevent and treat gastric cancer in relation to Tipα and nucleolin. Finally, the biological similarity between tumor promotion and the aging process is briefly discussed.

## 2. Tumor-Promoting Activities of Tipα Family Members, HP-MP1, and Tipα

TNF-α induces the clonal growth of v-Ha-*ras* transfected BALB/3T3 cells (Bhas 42 cells) as a model of initiated cells, whereas it does not induce the clonal growth of BALB/3T3 cells lacking v-H-*ras* [17]. Considering the functional similarity between TNF-α and HP-MP1, *hp-mp1* was transfected into Bhas 42 and BALB/3T3 cells, yielding Bhas/mp1 and BALB/mp1 clones. Bhas/mp1 showed strong induction of TNF-α, but BALB/mp1 did not. The results showed that transfection of *HP-MP1* into Bhas 42 cells strongly induced *tnf-α* expression in conjunction with v-H-*ras* [21]. The evidence was well supported by a review article stating that *H. pylori* infection stimulates the expression of various pro-inflammatory cytokines, including TNF-α, IL-8, IL-1α, IL-1β, and IL-6 in the gastric mucosa [23]. Although the biochemical role of v-Ha-*ras* in Bhas 42 cells is not discussed here, a recent paper provides important information that a KRAS splice variant, KRAS4A, directly regulates hexokinase 1 on the outer mitochondrial membrane, possibly in relation to the Warburg effect [24]. Bhas/mp1 clones formed tumors in 100% (18/18) of the injected sites by subcutaneous implantation within 20 days, whereas Bhas/ure clones, which were *urease B*-transfected Bhas42 cells, induced 33.3% (6/18) tumor development. These results showed that HP-MP1 had strong tumorigenicity compared with urease B and that HP-MP1 acts as a tumor promoter and induces human gastric cancer during *H. pylori* infection (Table 1) [21].

*HP0596* was found to have 94.3% homology with HP-MP1 based on an in silico database search of *H. pylori* strain 26695 and was named tipα [22]. Recombinant Tipα (rTipα) consisting of a His-tag and 172 amino acids was obtained by subcloning *HP0596* into the His-tag expression vector pET28(a)+. Treatment with rTipα protein stimulated *tnf-α* expression by approximately 26-fold in Bhas 42 cells, and 2.6 μM rTipα induced transformed foci with 18.0 foci/well of Bhas 42 cells (Table 1). This was similar to TPA (1.6 μM), which induces 38.0 foci/well, indicating that Tipα produced by *H. pylori* acts as a tumor promoter [22].

## 3. Secretion of Tipα from *H. pylori*

Western blot analysis with a specific anti-Tipα antibody identified two forms of Tipα in the culture medium. In the absence of dithiothreitol (DTT), Tipα was detected as a wide band at 38 kDa and a small band at 19 kDa. Similar Tipα expression patterns were found in extracts from various *H. pylori* strains, such as 26695Δ*cagPAI* (26695 strain with internal deletion within *cagPAI*), ATCC43504, SS1, and four *H. pylori* isolates from patients with gastritis, gastric ulcer, duodenal ulcer, and gastric cancer. All *H. pylori* strains produce Tipα, a homodimer of 38 kDa, and secrete it into the culture medium [22].

Tipα secretion was further examined using 28 *H. pylori* clinical isolates from 17 patients with gastric cancer and eleven patients with chronic gastritis. *H. pylori* isolates from these Japanese patients produced Tipα and CagA. To compare the amounts of Tipα in clinical isolates, 10 ng of secreted Tipα/10^9^ CFU/mL was considered as one relative unit. *H. pylori* isolates from cancer patients secreted 1.4–13.4 relative units of Tipα, whereas those from gastritis patients secreted 0.8–6.7 relative units (Figure 1). In addition, three of the eleven gastritis patients later developed gastric cancer and *H. pylori* isolated from them secreted higher amounts of Tipα, similar to those from cancer patients. This suggests that the secreted Tipα homodimer acts as a tumor promoter in the cancer microenvironment of the human stomach [25].

## 4. Cellular Response to rTipα

Tipα has two cysteine residues (Cys5 and Cys7) in the N-terminal region (Figure 2A). To understand the underlying molecular mechanism of Tipα, a deletion mutant of r*Tipα* (*rdel-Tipα*) was generated by deleting six amino acids (from Leu 2 to Cys 7). The molecular weight of rTipα was 42 kDa without DTT and 21 kDa with DTT; however, rdel-Tipα was 20 kDa, regardless of the presence of DTT. rTipα strongly induced *tnf-α* gene expression by approximately 26-fold relative to basal levels in Bhas 42 cells, whereas rdel-Tipα did not induce *tnf-α* gene expression even at a concentration of 100 μg/mL [22]. Furthermore, treatment of Bhas 42 cells and the mouse gastric cancer cell line MGT-40 with rTipα activated NF-κB by IκB degradation and nuclear translocation of the p65 subunit of NF-κB. Treatment with MG-132, the proteasome inhibitor, suppressed both nuclear translocation of p65 and rTipα-induced *tnf-α* expression [26]. However, rdel-Tipα does not have an obvious effect on the activation of NF-κB. Taken together, these results suggest that the active form of rTipα is a homodimer with disulfide bonds formed by cysteine residues. 

One of the hallmarks of *H. pylori* infection is the upregulation of gastric epithelial IL-8 expression, and *H. pylori* eradication downregulates the expression of cytokines and chemokines [29]. Treatment of MGT-40 cells with rTipα upregulated 120 genes (over 2-fold), among which five chemokine genes showed more than 10-fold upregulation, including *cxcl1*, *cxcl5*, *cxcl2*, *cxcl10*, and *ccl2*; moreover, *ccl7* was 5.8-fold upregulated [30]. rTipα stimulated the production of IL-1α and TNF-α from macrophages, and Tipα knockout significantly decreased *H. pylori* colonization in mice [31]. rTipα induced the expression of IL-1β, IL-8, and TNF-α to higher levels in the human gastric cancer cell line SGC7901 than in the human gastric epithelial cell line GES-1. After blocking NF-κB with pyrrolidine dithiocarbamate (PDTC), the SGC7901 cells did not show any increase in Tipα-induced IL-1β and TNF-α [32]. These results suggest that Tipα is a strong inducer of inflammatory cytokines and chemokines in *H. pylori* and is involved in tumor promotion and progression in the human stomach. 

## 5. Crystal Structures of Rdel-Tipα and N-Terminal Truncated rTipα 

Tipα has no amino acid sequence similarity to other *H. pylori* pathogenicity factors. As rTipα failed to form crystals, the stereochemical structure of rdel-Tipα was determined using multiple isomorphous replacement with anomalous scattering [27]. The rdel-Tipα monomer showed an elongated structure with an axis length of approximately 50 Å and a novel β1-α1-α2-β2-β3-α3-α4 topology (Figure 2A). A short helix formed with a flexible N-terminal region plays an important role for the dimerization of rdel-Tipα in a quaternary structure without a disulfide bridge. rdel-Tipα monomer A interacts with monomer B to form a heart-shaped dimer and exhibits a unique quaternary structure (Figure 3).

The first visible amino acid (Asp 19) was adjacent to monomers A and B. The structure of rdel-Tipα is maintained by the interaction of the N-terminal portions. The CD spectra of rTipα and rdel-Tipα shows similar features, which suggests that the structure of rTipα resembles that of rdel-Tipα in solution, although rTipα contains disulfide bonds and rdel-Tipα does not [27]. Thus, the activity of rdel-Tipα appears to be weak. Although there is no sequence homology between rdel-Tipα and any known proteins as per a sequence data search, Tsuge et al. reported that the lower part of rdel-Tipα is structurally homologous to dodecin by MArkovian TRAnsition of Structure evolution [27,33]. 

Tipα has attracted attention as a structurally novel protein. An N-terminal truncated version of Tipα (TipαN34) yielded two crystal structures with the same topology as that reported by Tsuge et al. [27,34]. The crystal structure of the Tipα monomer from *H. pylori* showed the presence of a mixed domain and helical domain, and the dimeric structure indicated a new scaffold protein for DNA binding [35]. Dimer forms of TipαN34 and Tipα are very similar to the stereochemical structure of del-Tipα (Figure 3). Surface plasmon resonance spectroscopy of the association between Tipα and DNA indicated that the affinity of rTipα for (dGdC)10 is 2400-fold stronger than that of rdel-Tipα [36]. Biochemical assays and molecular dynamic simulation of the DNA–Tipα interaction indicated that Tipα uses the dimeric interface as the DNA-binding site, and residues His60, Arg77, and Arg81 located at the interface are important for DNA binding [37]. These results raise questions regarding the interaction of *H. pylori* secreted Tipα with gastric epithelial cells.

## 6. Nucleolin as a Cell-Surface Receptor for rTipα

Specific binding of rTipα to MGT-40 cells using FITC-labeled rTipα (FITC-rTipα) showed the presence of a specific binding protein to the homodimer of FITC-rTipα on the surface of gastric cancer cells [25]. To characterize the binding protein for Tipα, rTipα-FLAG, and rdel-Tipα-FLAG constructs containing a six-histidine tag at the N-terminal region and a FLAG-tag (Asp-Tyr-Lys-Asp-Asp-Asp-Asp-Lys) at the C-terminal region were generated (Figure 2B). The biological activity of rTipα-FLAG and rdel-Tipα-FLAG were the same as that of rTipα and rdel-Tipα. A pull-down assay with anti-FLAG antibody detected 13 polypeptides that co-precipitated with rTipα-FLAG in the MGT-40 cell lysate but not with rdel-Tipα-FLAG. An 88 kDa polypeptide was identified as nucleolin, wherein a 40 kDa polypeptide was found to be a fragment of nucleolin as determined by LC–MS. Another polypeptide of less than 50 kDa was a ribosomal protein L4 fragment, and others remained unconfirmed. The 88 kDa polypeptide was confirmed to be nucleolin by immunoblotting with an anti-nucleolin antibody. Several 50–70 kDa polypeptides reacted with the anti-nucleolin antibody, suggesting degradative fragments of nucleolin. Additional experiments confirmed that rTipα directly binds to a His-tagged nucleolin fragment (284–710) containing four RNA-binding domains [28].

Nucleolin is involved in several biological functions, including gene expression, chromatin remodeling, DNA recombination and replication, mRNA stabilization, and apoptosis [38]. Fractionation studies of MGT-40 and human gastric cancer cells revealed that the amounts of full-size nucleolin were comparable in the membrane and nuclear fractions [39]. Importantly, nucleolin was not detected in the membrane fraction of the mouse normal glandular stomach, whereas nucleolin was present in the membrane of mouse gastric cancer cells, consistent with previous experiments (Figure 4). Nucleolin expression levels in the nuclear fraction were lower in normal mouse epithelial cells than in MGT-40 cells. Human gastric cancer cell lines also showed high amounts of nucleolin in membrane fraction [28]. These results suggest that localization of nucleolin in the membrane is important for Tipα-nucleolin complex formation during *H. pylori* infection [28].

Confocal laser scanning microscopy reveals the presence of fluorescent-rTipα spots in the nuclei of MGT-40 cells treated with rTipα, supporting the notion that rTipα is internalized into the nuclei [25]. Incubation of MGT-40 cells with anti-nucleolin 295 (anti-NUC295) antibody, which recognizes surface nucleolin, enhanced incorporation of rTipα into cells (Figure 5A). Anti-NUC295 antibody enhanced *tnf-α* gene expression induced by rTipα in a dose-dependent manner, whereas rabbit IgG- or anti-nucleolin H-250 antibody did not affect *tnf-α* gene expression (Figure 5B). IgG and anti-nucleolin H-250 antibodies do not recognize cell-surface nucleolin. The results indicated that the rTipα–nucleolin complex is internalized into MGT-40 cells and induces *tnf-α* gene expression [28]. Thus, we believe that anti-NUC295 stimulates complex internalization by its interaction with the surface receptor nucleolin.

## 7. Epithelial-Mesenchymal Transition (EMT) Induced by rTipα

EMT is recognized as the acquisition of mesenchymal cell phenotypes by epithelial cells. Cancer cells exhibit phenotypes similar to mesenchymal cells associated with metastatic states, involving morphological changes and gene expression, acquisition of cell motility, and invasiveness into adjacent normal tissues [40]. EMT is induced in epithelial cells by various stimuli, including TNF-α, tumor growth factor-β, hepatocyte growth factor, hypoxia-inducible factor 1α, and several transcription factors [40]. rTipα induces EMT in human gastric cancer cells and migration of human gastric cancer cell lines MKN-1, AGS, MKN-7, and MKN-28 under serum-free conditions, as determined by the Transwell assay. MKN-1 cells (adenosquamous cell carcinoma) treated with rTipα began to show an elongated cell morphology at 1 h, and most of the cells were elongated within 3 h under serum-free conditions. Nucleolin-targeted siRNA-n1 and siRNA-n2 dominantly downregulate membrane-localized nucleolin rather than that of cytosolic and nuclear fractions and inhibit migration of cells induced by rTipα, whereas the negative control siRNA-nc had no effect. These results indicated that binding of rTipα to surface nucleolin induces the migration and elongation of MKN-1 cells [41].

Atomic force microscopy measurements indicate that metastatic mouse B16-F10 cells with high motility have low stiffness, whereas low metastatic B16-F1 cells with low motility have high cell stiffness [42]. Treatment of MKN-1 cells with rTipα reduced Young’s modulus from 2703 Pa to 2065 Pa, indicating that rTipα induces lower cell stiffness and leads to higher cell motility. In addition, rTipα induced expression of vimentin, a marker of EMT, in MKN-1 cells, although rTipα did not show downregulation of E-cadherin. Thus, rTipα induces EMT in human gastric cancer MKN-1 cells [41].

Other investigators have reported similar results. A different recombinant Tipα (also described as rTipα in this manuscript) induced the morphological changes indicating EMT and stimulated the growth and motility of SGC7901 cells. rTipα activated IL-6/STAT3 signaling, and this effect was abolished by blocking the signaling pathway [43]. *H. pylori* antigenic Lpp20 is a lipoprotein localized on the external membrane of *H. pylori* and is secreted inside vesicles along with two other proteins, HP1454 and HP1457. The crystal structure of Lpp20 is similar to that of Tipα, and Lpp20 stimulates EMT, cell motility, and downregulation of E-cadherin in gastric cancer cells [44].

## 8. Inhibition of Tipα-Associated Gastric Carcinogenesis by BTG2

*btg2* is a human ortholog of TPA-inducible sequence 21 (*tis21*) [45]. *tis21* was first identified as one of the immediate early-response genes in mouse 3T3 fibroblasts treated with TPA [46]. *btg2* was cloned from a chromosomal rearrangement in B-cell chronic lymphocytic leukemia. The human antiproliferative *btg2* has strong sequence similarity to *tis21/pc3* [47]. *btg2^(tis21/pc3)^* is a p53 target gene that functions as a tumor suppressor and inhibits carcinogenesis in the thymus, prostate, kidney, and liver [48,49]. BTG2 is frequently lost in human cancers, whereas it is constitutively expressed in the epithelium and parietal cells of the gastric gland [50]. Although *H. pylori* infection upregulates BTG2 in the mucous epithelium, its expression is lost in human gastric adenocarcinoma. Adenovirus transduction of *btg2^tis21^* inhibited *tnf-α* expression induced by rTipα in human and mouse gastric cancer cell lines, MKN-1 and MGT-40. Furthermore, ectopic expression of *btg2^tis21^* inhibited the transcription of nucleolin by downregulating the binding of the transcription factor Sp1 to the promoter of the nucleolin-encoding gene *(NCL)*. Overexpression of *btg2^tis21^* significantly reduced nucleolin in the membrane fraction of cancer cells, and downregulation of *btg2^tis21^* increased nucleolin in gastric cancer tissues. High expression levels of *BTG2* and decreased nucleolin expression are associated with better overall survival in patients with poorly differentiated gastric cancer [50]. In summary, these findings suggest that BTG2^/TIS21^ downregulates nucleolin and facilitates the inhibition of carcinogenesis after *H. pylori* infection.

## 9. New Strategies to Prevent and Treat Gastric Cancer

The significance of *H. pylori* eradication with antibiotics for the prevention of gastric cancer is well recognized in Japan as well as by the International Agency for Research on Cancer (IARC) working group [51]. Based on the pathogenicity of Tipα, the effects of a prophylactic vaccine antigen were examined. C57BL/6 mice were immunized by administration of CpG, rTipα+CpG, and rdel-Tipα+CpG via the intranasal route. After 8 weeks, the mice were inoculated with *H. pylori*, and the number of colonizing *H. pylori* in the stomach and the histological damage due to gastritis were evaluated. These results suggest that immunization with rTipα and rdel-Tipα confers a protective effect against *H. pylori* infection [52], thereby highlighting an excellent strategy for vaccination using non-pathogenic rdel-Tipα. Furthermore, the presence of serum antibody against Tipα was associated with the increase in the chances of *H. pylori* infections, and subjects with serum antibodies against Tipα, CagA, and HP0175 carried an increased risk of atrophic gastritis [53]. Future studies need to be conducted to determine whether serum antibodies against Tipα are associated with the risk of gastric cancer. 

Cell-surface nucleolin functions as a receptor for various ligands, including lactoferrin, endostatin, midkine, and human immunodeficiency virus (HIV). Anti-HIV pseudopeptide (HB-19) binds to surface nucleolin and exhibits anti-carcinogenic activity in vivo [54]. HB-19 inhibited the growth of breast cancer in an athymic nude mouse model and the development of spontaneous melanoma in RET transgenic mice [55]. AS1411 is a well-investigated anticancer DNA aptamer (26-mer) and specifically binds to nucleolin on the cell surface, resulting in inhibition of nucleolin function in vitro and in vivo. Treatment with AS1411 inhibited the growth of human gastric cancer cell lines MKN-45 and MNK-1 by inducing the S-phase cell cycle [39]. Furthermore, AS1411 inhibited the binding of FITC-labeled rTipα to MKN-1 cells, resulting in the inhibition of migration induced by rTipα. Lactoferrin, another ligand of nucleolin, showed similar inhibitory effects as AS1411 [39]. These results indicate that cell-surface nucelolin is a promising therapeutic target for gastric cancer. The role of surface nucleolin as a mediator for carcinogenic, anti-carcinogenic, and disease-related ligands has been recently reviewed [56].

## 10. Discussion

Clinical and epidemiological studies on various strains of *H. pylori* revealed that American strains are tightly linked to gastric cancer, whereas African strains are related to gastritis. Tipα was present in clinical isolates of Japanese patients with gastric cancer. It has been reported that American and Euro-Asian strains contain Tipα, whereas Africa1, Africa2, and East Asian strains do not harbor it [9]. These data suggest that Tipα plays a significant role in *H. pylori* gastric carcinogenesis. Evidence indicates that surface nucleolin is a carcinogenic receptor for Tipα, and the complex of Tipα and nucleolin is internalized into cells and stimulates tumor promotion and progression in human gastric cancer. 

CagA and VacA proteins are well-known virulence factors associated with gastric cancer. CagA protein is injected into target cells by the Cag type IV secretion system encoded by *cagPAI* [57]. CagA protein may bind to the inner membrane, where Src family kinases phosphorylate its tyrosine residues. Phosphorylated and unphosphorylated CagA interact with various proteins and activate signaling pathways in the host cells, resulting in the induction of IL-8 and destruction of cell polarity. The VacA protein is secreted by *H. pylori* via a type V autotransport secretion system [58]. VacA induces vacuole formation in target cells and induces necrosis and apoptosis of cells. The prevalence of infections with strains expressing CagA and VacA is high (78%), regardless of the pathological status in the gastroduodenum of patients in Japan, indicating the presence of an additional factor involved in gastric cancer development [59]. Specifically, amounts of secreted Tipα are different among clinical isolates derived from patients with gastritis and gastric cancer in Japan, although how Tipα is secreted from *H. pylori* is unknown. Tipα induces strong expression of TNF-α, IL-8, IL-1, and chemokines and promotes EMT and tumor development in cells via mechanisms different from those of CagA and VacA. Due to the uniqueness of its mechanism and structure, we believe Tipα is involved in gastric cancer in conjunction with CagA and VacA.

A recent study on age-associated chronic inflammation (inflammaging) has attracted considerable attention. Plasma from old (28–29 months) mice contained higher levels of pro-inflammatory cytokines (TNF-α and IL-6), anti-inflammatory cytokines (IL-4), and chemokines and growth factors (CSF1) than plasma from young (3 months) mice. Moreover, the levels of pro- and anti-inflammatory cytokines were increased in the conditioned medium of cultured primary fibroblasts from the ears of old mice [60]. This suggests that increased levels of inflammatory cytokines secreted from fibroblasts partly reflect the aging process because aged individuals have high levels of inflammatory cytokines. Since aging is similar to tumor promotion and progression in the human cancer microenvironment, strategies to slow down the aging process might overlap with cancer prevention in humans.

## Figures and Tables

**Figure 1 toxins-13-00181-f001:**
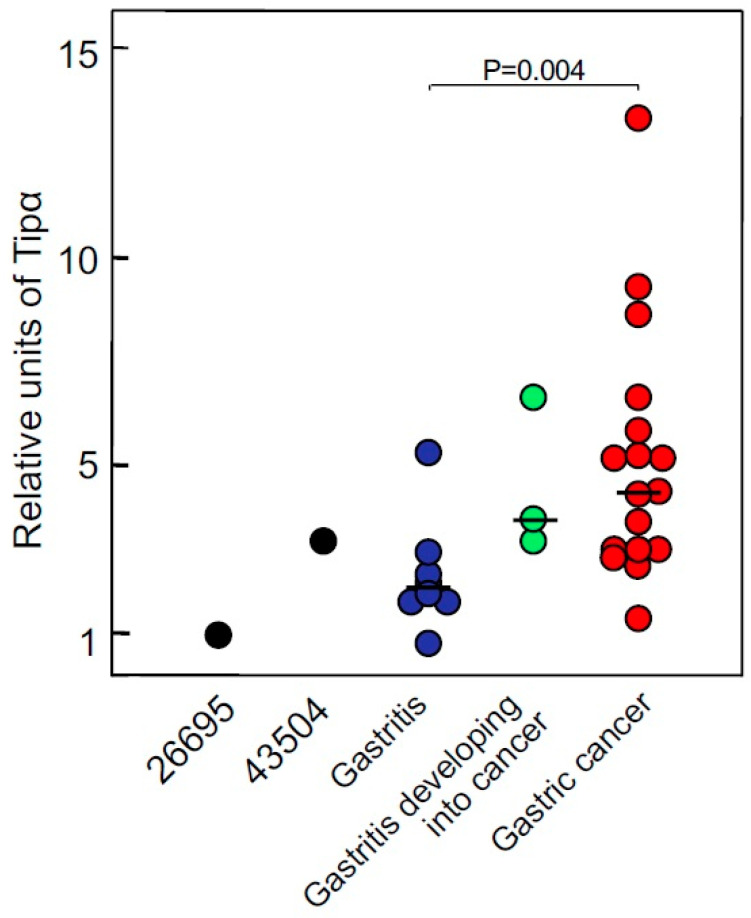
Relative units of Tipα secreted from *H. pylori* strains and clinical isolates. Two *H. pylori* strains (black), *H. pylori* isolated from patients with gastritis (blue), gastritis developing into cancer (green), and gastric cancer (red). *H. pylori* isolated from patients with cancer secreted 1.4–13.4 relative units of Tipα, and that from patients with gastritis secreted 0.8–6.7 relative units [25].

**Figure 2 toxins-13-00181-f002:**
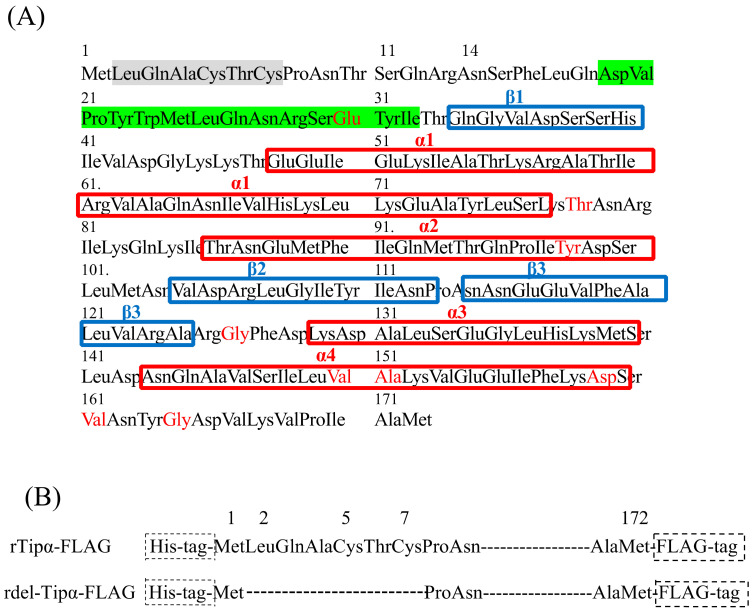
Primary structure of Tipα (**A**) and partial sequence of rTipα-FLAG and rdel-Tipα-FLAG (**B**). (**A**) Red characters in the sequence indicate different amino acids between HP-MP1 and Tipα. The secondary structures are shown in red (α-helix) and blue (β-sheet), and the N-terminal flexible region is shown in green. Six amino acid indicated in shadow are deleted in rdel-Tipα [27] (**B**) rTipα-FLAG and rdel-Tipα-FLAG, which have a His-tag at the N-terminal region and a FLAG-tag (Asp-Tyr-Lys-Asp-Asp-Asp-Asp-Lys) at the C-terminal region, respectively. rdel-Tipα-FLAG has a deletion of six amino acids (from Leu 2 to Cys 7 containing two cysteine residues at Cys 5 and Cys 7) [28].

**Figure 3 toxins-13-00181-f003:**
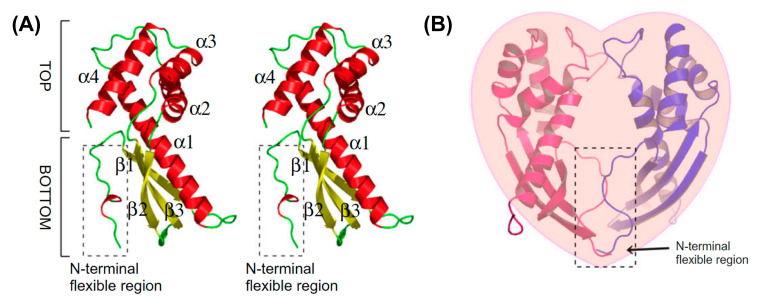
Tertiary structure of rdel-Tipα. (**A**) Stereochemical structure of rdel-Tipα monomer, (**B**) heart-shape structure of rdel-Tipα dimer [27].

**Figure 4 toxins-13-00181-f004:**
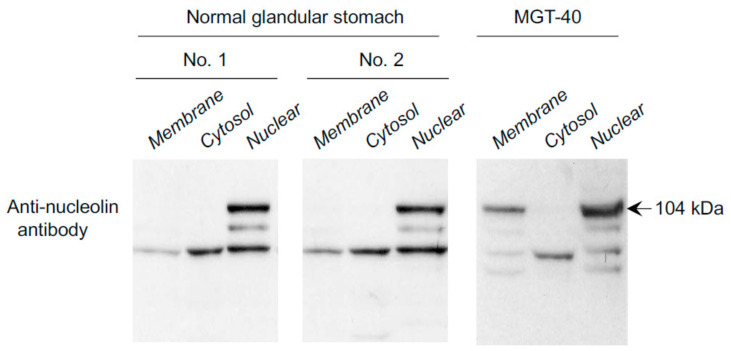
Subcellular localization of nucleolin in the epithelial cells of the mouse normal glandular stomach and in mouse gastric cancer MGT-40 cells. Nucleolin was not detected in the membrane of epithelial cells of the mouse normal glandular stomach, whereas nucleolin was detected in the membrane of MGT-40 cells [39].

**Figure 5 toxins-13-00181-f005:**
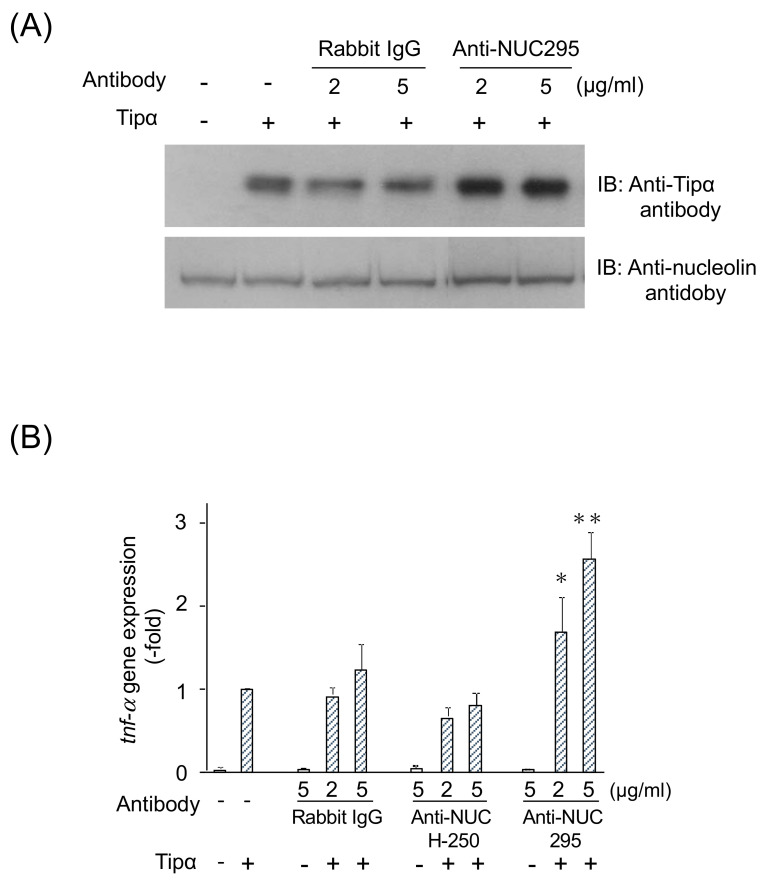
Enhancement of rTipα incorporation into MGT-40 cells induced by anti-NUC295 antibody and *tnf-α* gene expression induced by rTipα. (**A**) Incorporation of rTipα was confirmed by Western blotting with an anti-rTipα antibody. Nucleolin levels were determined using an anti-nucleolin antibody. IB: immunoblotting. (**B**) Relative *tnf-α* gene expression is shown as fold change compared with that of cells treated with 50 μg/mL rTipα after normalization to *gaphd* gene expression levels [28].

**Table 1 toxins-13-00181-t001:** Tumor-promotion activity of HP-MP1 and Tipα.

	Bhas 42 Cells (with *v-H-ras*)		BALB/3T3 Cells	
	*tnf-α* Gene Expression	Tumorigenicity/ Transformation	*tnf-α* Gene Expression	Tumorigenicity/ Transformation
Transfection	*(Bhas/mp1)*		*(BALB/mp1)*	
*hp-mp1* gene	High	High	Very low	No
	*(Bhas/ure)*		*(BALB/ure)*	
*urease B* gene	Low	Low	Very low	No
Treatment				
rTipα protein	High	High	Low	No
rdel-Tipα protein	Low	-	No	No

## Data Availability

The datasets generated and/or analysed during the current study are available from the corresponding author on reasonable request.

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
