# Peer review of "Role of TNF-α-Inducing Protein Secreted by Helicobacter pylori as a Tumor Promoter in Gastric Cancer and Emerging Preventive Strategies"

_toxins, 2021, doi:10.3390/toxins13030181_

Round 1
Reviewer 1 Report
This is an interesting and up-to-date review on the role of a protein produced by H. pylori (i.e. Tipalpha) as a tumor promoter in gastric carcinogenesis.
The manuscript is well written, the purpose of the review is well defined, and the data are presented in a clear manner. In my opinion the manuscript is informative and relevant both on basic science and a clinical level, indicating a novel putative molecular pathway that might serve as a target for gastric cancer therapy
Author Response
Thank you for the favourable response. We are pleased to receive your encouraging comments.

Reviewer 2 Report
This manuscript reviews Tipα and HP-MP1, which is secreted by H. pylori, on several points of view such as follows tumor promoting activities, cellular response, structure, receptor, epithelial-mesenchymal transition, inhibition mechanisms by BTG2/TIS21. And the authors introduce new strategies for the prevention and treatment of gastric cancer.
I think that the topic is important and valuable for reader.
comments
There are numerous genome sequences of the H. pylori on data base.
If there are any reports about gene comparison analysis or molecular evolution of Tipα family in H. pylori, please add it in this article.
Are there any references about serum antibody responses to Tipα in the patient infected with H. pylori. Is the occurrence of gastric cancer correlate to the seropositivity of anti-Tipα?
Author Response
Thank you very much for your favourable and pertinent comments. Based on your comments, we have revised the manuscript, and included a reference related to serum antibodies against Tipα. We hope that you find that these revisions to be sufficient and that they have strengthened our manuscript.
Response: We appreciate your constructive comment. There are no published articles reporting gene comparison analyses or evolution of the Tipα family in H. pylori. Thus, we conducted a preliminary evolution analysis of the Tipαfamily as shown below. Due to time constraints, we did not include this preliminary result in the revised manuscript. After completing the analysis, we would like to submit this data as a part of an upcoming manuscript.
Response : We added relevant text from a paper titled “Multiplex serology of Helicobacter pylori antigens in detection of current infection and atrophic gastritis – A simple and cost-efficient method” (Shafaie E et al. Microb Pathog 199:137-144, 2018), and added the sentences on page 9 as follows: Furthermore, the presence of serum antibody against Tipα was associated with the increase of the chances of H. pylori infections, and subjects with serum antibodies against Tipα, CagA, and HP0175 carried an increased risk of atrophic gastritis [53]. Future studies need to be conducted to determine whether serum antibodies against Tipα are associated with the risk of gastric cancer.

Reviewer 3 Report
This manuscript by "without name" describe interesting data on the role of TNF-Alpha inducing protein secreted by Hpylori.
This review suffer from a very important quantity of data that could benefit from numerous modifications.
Some part are redundant and deserve revision (since the first paragraph of the introduction). Authors have to deeply revise the manuscript.
Genes have to be italicized and be written without capital letter. "et al." has to be written in italic. Moreover, number below 12 have to be written in full letters.
Scheme that summarize paragraph 2 could be of interest.
A crystal (or proteomic) representation of the Tipa/rdel-Tipa and Nterm-trunc Tipa could be of interest to understand the 3D representation
Author Response
Thank you for your excellent comments. Based on your comments, we have revised the manuscript as follows:
Response: We shorten the first paragraph of the introduction. Deleted parts are indicated in red characters as follows:
Helicobacter pylori is a gram-negative bacterium that resides in the gastric lumen and is an important human pathogen. In 1984, Marshall and Warren published their findings on the association of H. pylori infection with chronic gastritis and peptic ulcers [1]. H. pylori was later classified as a definitive carcinogen in humans (Class 1) based on epidemiological studies [2]. Determination of The complete genome sequence of H. pylori strain 26695, which comprises of 1,667,867 base pairs, revealed the presence of well-developed harbors intricate systems responsible for motility, iron uptake, and DNA restriction and modification [3]. Infection with H. pylori strains derived from humans experimentally induced gastric cancer in Mongolian gerbils [4,5]. Administration of the carcinogen N-methyl-N’-nitro-N-nitrosoguanidine (MNNG), as a tumor initiator, followed by H. pyloriinoculation, as tumor promoter, induced adenocarcinoma in the glandular stomach of Mongolian gerbils [6]. The results showed that H. pylori exhibits both tumor-initiating and tumor-promoting activities. The Nobel Prize in Physiology or Medicine, 2005 was awarded jointly to Marshall and Warren “for their discovery of the bacterium H. pylori and its role in gastritis and peptic ulcer disease.” Epidemiological studies have revealed that H. pylori is relatively non-malignant in indiciduals of African ancestry, whereas it is harmful in humans of American ancestry. This indicates that all isolated H. pylori strains contain the genetic signatures of multiple ancestries [7,8]. Investigators determined The complete genome sequence of 60 H. pylori strains with origins spanning five continents, and were assigned the genomes to populations from America, Africa1, Africa2, EuroAsia, and EastAsia based on discriminant analysis of principal components. The vacuolating cytotoxin autotransporter (vacA) gene, which is associated with H. pylori pathogenicity, was identified in EuroAsian and Africa1 strains. Several vacA-like genes were found in Africa2 strains, but not in EastAsian and American populations [9]. The prevalence of cytotoxin-associated gene A (cagA)-positive strains is associated with a high incidence of atrophic gastritis and gastric cancer in the Japanese and Korean populations [10,11]. In addition, the cag pathogenicity island (cag PAI) is present in 100% of EastAsian and 60-70% of Western strains [12]. However, the genetic diversity of H. pylori genomes suggests that existence of other genes and proteins that contribute to the development of human gastric cancer [13]. This review summarizes a new gene family, tumor necrosis factor-α (TNF-α)-inducing protein (tipα), which is secreted by H. pylori and acts as a tumor promoter. The Tipα gene family includes Tipα, H. pylori membrane protein 1 (hp-mp1), and possibly jhp0543 [14].
Response: Thank you for your suggestions. We have noted the points highlighted by you and have made revisions accordingly: genes are italicized without capital letter and et al. is written in italics in the revised manuscript.
We would like to point out that we in case of numbers less than 12, we have written out them in full, except when fractions or decimals are involved or when they are accompanied by a unit, so as to optimize readability of the text.
Response: Thank you for this suggestion. According to your suggestion, we have added Table 1 summarizing paragraph 2 on page 3.
Response: We added 3D structure of del-Tipα in Figure 3 and the following sentence on page 6: Dimer forms of TipαN34 and Tipα are very similar to the stereochemical structure of del-Tipα (Figure 3).

Round 2
Reviewer 3 Report
.